# Heteropolyacids@Silica Heterogeneous Catalysts to Produce Solketal from Glycerol Acetalization

**DOI:** 10.3390/nano14090733

**Published:** 2024-04-23

**Authors:** Catarina N. Dias, Isabel C. M. S. Santos-Vieira, Carlos R. Gomes, Fátima Mirante, Salete S. Balula

**Affiliations:** 1LAQV/REQUIMTE—Laboratório Associado para a Química Verde e Departamento de Química e Bioquímica & Faculdade de Ciências, Universidade do Porto, 4169-007 Porto, Portugal; 2CICECO—Aveiro Institute of Materials, Department of Chemistry, University of Aveiro, 3810-193 Aveiro, Portugal; 3CIIMAR—Centro Interdisciplinar de Investigação Marinha e Ambiental & Faculdade de Ciências, Universidade do Porto, 4169-007 Porto, Portugal

**Keywords:** acetalization, solketal, heteropolyacids, mesoporous silica

## Abstract

The composites of heteropolyacids (H_3_PW_12_, H_3_PMo_12_) incorporated into amine-functionalized silica materials were used for the first time as heterogeneous catalysts in the valorization of glycerol (a major waste from the biodiesel industry) through acetalization reaction with acetone. The polyoxotungstate catalyst H_3_PW_12_@*Aptes*SBA-15 exhibited higher catalytic efficiency than the phosphomolybdate, achieving 97% conversion and 97% of solketal selectivity, after 60 min at 25 °C, or 91% glycerol conversion and the same selectivity, after 5 min, performing the reaction at 60 °C. A correlation between catalytic performance and catalyst acidity is presented here. Furthermore, the stability of the solid catalyst was investigated and discussed.

## 1. Introduction

The use of fossil fuels as an energy resource has led to increased environmental concerns due to the emission of greenhouse gases, known to be one of the main contributors to climate change [1,2]. The growing environmental awareness has led to the implementation of legislation predicting a decrease in fossil fuel consumption. The interest in new energy alternatives has peaked, with particular attention being paid to biomass and its application in fuel formulation [3]. 

Biodiesel production has increased in recent years, leading to a large amount of glycerol waste, a known by-product obtained in biodiesel formulation. Biodiesel production results in the formation of glycerol at 10%, with purity ranging from 50 to 55% [1]. The market is unable to utilize this excess of glycerol since it cannot be utilized in its crude state, and the purification process can become extremely costly when performed on a large scale [4,5]. However, it is essential to transform glycerol into compounds of high commercial value due to its outstanding potential as a raw material for generating value-added products [3,6]. Various catalytic reactions for glycerol valorization have been studied, with acetalization gaining exponential interest, as the products obtained from this reaction can contribute to a circular economy model associated with the biodiesel industry [1]. Acetalization of glycerol in the presence of carbonyl compounds (aldehydes or ketones) yields solketal (2,2-dimethyl-1,3-dioxolone-4-methanol), acetal (2,2-dimethyl-1,3-dioxan-5-ol), and water. Solketal is a promising compound, thanks to its extraordinary performance as an oxygenated fuel additive, improving fuel characteristics [5]. Its production can be favored by using acid catalysts, such as Brönsted or Lewis acids [6,7]. Keggin heteropolyacids (HPAs) are an attractive group of acid catalysts, highly active in oxidative processes or acid-catalyzed reactions, thanks to the extensive number of active sites available on their surface [8,9,10]. The most studied HPAs are the phosphotungstic acid (H_3_PW_12_O_40_, abbreviated as H_3_PW_12_), phosphomolybdic acid (H_3_PMo_12_O_40_, abbreviated as H_3_PMo_12_), and silicotungstic acid (H_4_SiW_12_O_40_, abbreviated as H_4_SiW_12_) [11]. Da Silva et al. studied the influence of different HPAs on glycerol conversion in the presence of propanone [12]. In particular, H_3_PW_12_O_40_ distinguished itself, obtaining higher conversions (83%, at room temperature after 2 h) than other heteropolyacids [12]. Similarly, Julião et al. reported the activity of HPAs in acetalization reactions at room temperature, observing the following behavior: PW_12_ (99.2%) > PMo_12_ (91.4%) > SiW_12_ (90.7%), after only 5 min of reaction [13]. Da Silva et al. demonstrated that the Sn_2_SiW_12_O_40_ salt is more active (>99% conversion, after 1 h of reaction, at room temperature) than its tin (II) chloride salt precursor, as well as other heteropoly salts and solid-supported catalysts [4].

Despite the efficiency that HPAs have demonstrated in acetalization reactions, the recycling capacity of these catalysts is rarely reported. This type of catalyst is soluble in polar solvents, a feature that hinders catalyst recovery and reuse, being an obstacle in continuous solketal production [14]. One feasible method to solve this problem is to support HPAs on inert support materials, such as mesoporous silicas [15], MOFs [16], active carbon [17], or ion-exchange resins [18]. In particular, mesoporous silica-based catalysts have acquired attention due to the stability of these supports and the high diffusion facilitated by their tunable pore structure [19]. Chen et al. investigated the catalytic efficiency of a cesium salt of phosphotungstate (Cs_2.5_) both in bulk and supported in KIT-6 silica [20]. Both catalysts achieved similar conversions, with the reaction time being shortened (from 60 min to 15 min) when using the supported material. Furthermore, the Cs_2.5_/KIT-6 catalyst was reused for three cycles with no loss of activity [20]. Castanheiro reported the use of silica, supporting HPAs for the acetalization of glycerol with citral, where H_3_PW_12_@KIT-6 showed high stability after five consecutive cycles [21].

In this work, homogeneous HPAs and heterogeneous HPAs, supported on functionalized mesoporous SBA-15 silica, were applied in the acetalization reaction of glycerol with acetone. Two different heterogeneous catalysts were prepared by immobilizing H_3_PW_12_ and H_3_PMo_12_ in amine-functionalized silica support (*Aptes*SBA-15). Electrostatic interactions between HPAs and functionalized *Aptes*SBA-15 will promote an efficient immobilization of HPAs active catalytic centers. The influence of various reaction parameters, such as catalyst acidity and temperature, was analyzed to improve the conversion and selectivity of solketal as the product of interest. Catalyst stability was also investigated.

## 2. Materials and Methods

### 2.1. Materials

All reagents and solvents used in this work were purchased from commercial sources and used without further purification: Glycerol (99.92%, VWR Chemicals, Lisbon, Portugal), acetone (99.5%, Honeywell, NC, USA), methanol (99.9%, Fisher Scientific, Loughborough, UK), Pluronic P123 (Aldrich, London, UK), hydrochloric acid (HCl, 37%, Aldrich, London, UK), tetraethoxysilane (TEOS, Aldrich, London, UK), 3-aminopropyltriethoxysilane (*Aptes*, 98%, Aldrich, London, UK), anhydrous toluene (99.8%, Aldrich, London, UK), acetonitrile (Carlo Reba Reagents, Milano, Italy), ethanol (99.8%, Honeywell, NC, USA), phosphomolybdic acid hydrate (H_3_[PMo_12_O_40_].nH_2_O, Aldrich, London, UK), phosphotungstic acid hydrate (H_3_[PW_12_O_40_].nH_2_O, Aldrich, London, UK), and silicotungstic acid hydrate (H_4_[SiW_12_O_40_].nH_2_O, Aldrich, London, UK).

### 2.2. Preparation of Catalysts

The SBA-15 solid support was prepared through a hydrothermal synthesis procedure adapted from the literature [15]. Pluronic P123 (2.0 g) was dissolved in aqueous HCl (2 M, 60 mL) and distilled water (15 mL) under stirring at 40 °C, and then TEOS (4.4 g) was added dropwise. The mixture was stirred for 24 h at 40 °C. After being cooled, the mixture was transferred to a Teflon autoclave, and the temperature was raised to 100 °C for another 24 h. The resulting precipitate was filtered, dried, and calcinated at 550 °C for 5 h with a ramp of 1 °C min^−1^.

The surface of SBA-15 was functionalized via a post-grafting methodology [15]. The SBA-15 support (1 g) was activated for 2 hours at 100 °C, under vacuum. Then, it was dispersed in anhydrous toluene (60 mL) and *Aptes* (0.7 mL). The mixture was placed in an N_2_ atmosphere, under constant stirring at 90 °C for 24 h. The resulting material was filtered, washed with toluene and ethanol, and dried under vacuum at 80 °C for 2 h.

The HPAs@*Aptes*SBA-15 composites were prepared through an impregnation technique adapted from the literature [22]. The functionalized support *Aptes*SBA-15 (0.3 g) and acetonitrile (12 mL) were dispersed under constant agitation at room temperature. Then, the HPA (0.9 g) was added, and the mixture was stirred for 72 h, at room temperature. The obtained material was filtered, washed with acetonitrile and ethanol, and dried.

### 2.3. Catalyst Characterization

FTIR-ATR spectra were recorded on a Perkin Elmer Spectrum BX spectrometer (Perkin Elmer, MA, USA), using the ATR operation mode. The spectra were acquired in the 400–4000 cm^−1^ region, with a resolution of 4 cm^−1^ and 64 scans. All the representations shown in this work present arbitrary units of transmittance. Powder X-ray diffraction (XRD) patterns were collected at room temperature on a Rigaku’s Smartlab diffractometer (Rigaku, Tokyo, Japan) operating with a Cu radiation source (λ1 = 1.5405980 Å; λ2 = 1.5444260 Å; λ2/λ1 = 0.500) and in a Bragg–Brentano θ/2θ configuration (45 kV, 40 mA). XRD analyses were performed at the University of Aveiro—CICECO. Zeta potential measurements were performed at 25 °C in a Malvern Zetasizer Nano ZS spectrometer (Malvern Panalytical Ltd., Worcestershire, UK). All measurements were repeated three times to verify the reproducibility of the results, and all the samples were prepared by dissolving 1.0 mg of each sample in Millipore water (2 mL). Scanning electron microscopy (SEM) and energy dispersive X-ray spectroscopy (EDS) studies were performed at “Centro de Materiais da Universidade do Porto” (CEMUP, Porto, Portugal) using an FEI Quanta 400 FEG ESEM electron microscope with an EDAX Genesis X4M energy dispersive X-ray spectrometer (Gatan, CA, USA), operating at 15 kV. The samples were studied as powders and were coated with an Au/Pd thin film by sputtering using the SPI Module Sputter Coater equipment. Nitrogen adsorption–desorption isotherms were measured at −186 °C, using AutoSorb equipment (Anton Paar QuantaTec, FL, USA). Samples were previously evacuated in situ under a high vacuum (10^−7^ bar) for 12 h at 100 °C. The surface area was calculated using the Brunauer–Emmett–Teller (BET) model. The pore volume was obtained using Barret–Joymer–Hallenda (BJH) calculations. HPAs quantification in the composites was possible using a Varian 820-MS spectrometer (Varian, CA, USA) through the quantification of molybdenum (Mo) and tungsten (W) present in the materials. Analyses were performed at the University of Santiago de Compostela, Spain. Elemental analysis of C, N, and H in *Aptes*SBA-15 was obtained using a Leco CHNS-932 elemental analyzer (LECO Corporation, St. Joseph, MI, USA). Analyses were performed at the University of Santiago de Compostela, Spain. The acidity strength of HPAs and HPAs@*Aptes*SBA-15 materials was obtained through potentiometric titration measurements performed in a TitraLab AT1000 Series instrument (Hach Company, Ames, IA, USA), using NaOH (0.025 M) as the base. Solutions of the materials and a NaCl solution were prepared in a 1:1 ratio and left at room temperature under stirring for 24 h. The suspension was separated by filtration.

### 2.4. Catalytic Experiments

Glycerol acetalization reactions with acetone, under a solvent-free system, were carried out in a closed borosilicate 5 mL catalytic reactor equipped with a magnetic stirrer and immersed in a thermostatically controlled liquid paraffin bath. The reactor was filled with the appropriate glycerol/acetone ratio (1:15) and left under agitation at the chosen temperature (25 °C or 60 °C) for 10 min to ensure homogeneity. Catalysts were then added, and the reaction was initiated. Reaction progression was evaluated in a Varian CP-3380 gas chromatograph (Varian, CA, USA), with a Suprawax-280 capillary column (Teknokroma Analítica SA, Barcelona, Spain) (30 m length, 0.25 mm internal diameter, and 0.25 µm film thickness), using H_2_ as the carrier gas (flow rate of 55 cm^3^/s). At least three repeated reactions were performed, and the error obtained was equal, or inferior to, 5% of the conversion of glycerol. 

## 3. Results and Discussion

### 3.1. Catalysts Characterization

The composite materials were prepared using an impregnation method involving the incorporation of heteropolyacids (HPAs) in amine-functionalized SBA-15 support (*Aptes*SBA-15). The support, functionalized support, and composites were characterized using several techniques, including elemental analysis, zeta potential (ζ), FTIR-ATR, powder XRD, nitrogen adsorption–desorption isotherms, ICP-OES, and SEM/EDS.

The SBA-15 and *Aptes*SBA-15 support presented zeta potentials of −17.5 mV and 29.3 mV, respectively. These values were consistent with the literature, where the change of signal (from negative to positive) has been reported to occur due to the successful functionalization of the amino groups on the support surface [23]. Elemental analysis of *Aptes*SBA-15 further confirmed the surface functionalization, yielding 2.4% nitrogen, corresponding to 1.1 mmol of *Aptes* per g of material. The FTIR-ATR spectra of the prepared materials are shown in Figure 1A, where the characteristic vibration modes of the silica support can be observed in the region 1100–400 cm^−1^, namely for the ν_as_ (Si-O-Si), ν_s_ (Si-O-Si), and δ (Si-O-Si) vibrations [24]. The presence of amino groups in the 1200–1000 cm^−1^ region could not be identified due to overlap with the siliceous support vibrational modes [25]. Furthermore, the characteristic band around 1510 cm^−1^, corresponding to N–H stretching vibrations, attributed to aminopropyl functionalization groups, could not be distinguished when comparing SBA-15 and *Aptes*SBA-15 spectra [26]. This is probably due to the low amount of *Aptes* on the surface of SBA-15. The incorporation of HPAs in the prepared composites was confirmed through the identification of the vibration bands at 1000–960 cm^−1^ and 800–760 cm^−1^, corresponding to ν_as_ (M=O_d_) and ν_as_ (M-O_c_-M) vibration modes, respectively [26]. Furthermore, the incorporation of HPAs in the composites was confirmed through ICP-OES analysis where loadings of 159 and 188 µmol of HPA per g of the composite were obtained for H_3_PW_12_@*Aptes*SBA-15 and H_3_PMo_12_@*Aptes*SBA-15, respectively. 

The XRD patterns of the materials are depicted in Figure 1B. The SBA-15 support has a hexagonal structure, with three characteristic well-resolved peaks in the low-angle area. The strong peak around 1° can be attributed to the (100) plane, and the two weak peaks around 1.5° and 1.8° to the (110) and (200) planes [27]. The functionalization of the surface of SBA-15 did not compromise the structure and crystallinity of the material, as *Aptes*SBA-15 exhibited the same peaks as the non-functionalized material. In the composites, the peaks of the (110) and (200) reflections were shifted to higher 2θ, as reported before in the literature for POMs@SBA-15 composites (POMs is the abbreviation for polyoxometalates) [22,24]. These results suggest that the structure of *Aptes*SBA-15 support was intact after HPA immobilization. The porosity of the composite materials was measured through N_2_ adsorption–desorption experiments. Both composites exhibited a type IV isotherm with H1 hysteresis loops (Appendix A), characteristic results obtained for mesoporous materials [24,28]. The textural properties of the composites and the supports are described in Table 1. It is possible to observe that the pore volume and surface area are steadily decreasing in the following order: HPA-containing composites < aminopropyl functionalized SBA-15 < SBA-15. This indicates the successful immobilization of HPAs in porous *Aptes*SBA-15, as well as an effective functionalization of *Aptes* groups in the SBA-15 support. The morphology of the prepared *Aptes*SBA-15 materials was evaluated through SEM analysis (Appendix A), where the characteristic hexagonal elongated particles of the SBA-15 support were observed [24,28]. After HPA’s incorporation, it was observed that the morphology of the silica support was maintained (Figure 2). The presence of HPA in the composite was further confirmed by the presence of the molybdenum and tungsten in EDS analysis (Figure 2).

### 3.2. Acidity Characterization

The acidity present in HPAs and HPAs@*Aptes*SBA-15 composites was evaluated through potentiometric titration, and the results are shown in Table 2. Per gram of material, the HPAs present higher acidity than the corresponding HPAs@*Aptes*SBA-15 composites. The decrease in acidity is related to the interaction of HPA with the aminopropyl groups, leading to the formation of an ammonium salt (where the proton of the HPA is less accessible). Further, the concentration of released H^+^ ions was determined to be 3.36 and 0.47 mmol/g for H_3_PMo_12_@*Aptes*SBA-15 and H_3_PW_12_@*Aptes*SBA-15, respectively. 

### 3.3. Catalytic Activity of HPAs

Previous work developed in the group studied the influence of experimental conditions on the acetalization reaction of glycerol with acetone using homogeneous HPAs as catalysts [13]. Optimized results were obtained with a ratio of 1:15 (glycerol/acetone), 3 wt% of HPA (in comparison to glycerol weight) at 25 °C. In this work, the influence of catalyst acidity and molarity was investigated using homogeneous HPAs in the acetalization of glycerol with acetone. Reaction parameters (acetone/glycerol ratio and temperature) were maintained using the optimized parameters previously reported by Julião et al. [13]. The catalyst acidity of 0.373 mmol H^+^/g was used for the different HPA catalysts (Table 3). The three HPAs (H_3_PW_12_, H_3_PMo_12_, and H_4_SiW_12_) were studied at 25 and 60 °C. The results obtained are shown in Figure 3. 

At 25 °C, the H_3_PW_12_ and H_3_PMo_12_ catalysts distinguished themselves, reaching 97% conversion after 30 and 60 min, respectively. The H_4_SiW_12_ exhibited lower conversion, reaching 84% after 60 min. At 60 °C, all three HPAs showed slightly lower conversions than those at 25 °C, with H_3_PW_12_ exhibiting the best results (92% conversion after 30 min). Solketal selectivity was above 97% for all catalysts, and only a vestigial 2,2-dimethyl-1,3-dioxan-5-ol amount was observed. As such, for the three HPAs, the optimized reaction temperature was 25 °C, as previously reported in the literature [4,13,20]. Therefore, maintaining the acidity of the catalyst, the catalytic activity follows the order: H_3_PW_12_ > H_3_PMo_12_ > H_4_SiW_12_. Previously, our research group performed the same catalytic reactions but maintained the amount of catalyst (3 wt% of HPA compared to glycerol weight, i.e., 57 µmol of HPA) [13]. However, the results obtained for glycerol conversion were identical to the present work, mainly using H_3_PW_12_ and H_3_PMo_12_ catalysts (Appendix A). An important advantage in the present work is that, with an acidity normalization of 0.373 mmol H^+^/g, a much lower amount of catalyst was needed (1.7 µmol of the most active H_3_PW_12_, compared to the previous 57 µmol required).

### 3.4. HPAs@AptesSBA-15 Catalytic Performance

The most active HPAs for the acetalization of glycerol were immobilized in an amine-functionalized *Aptes*SBA-15 support, and the composites (H_3_PW_12_@*Aptes*SBA-15 and H_3_PMo_12_@*Aptes*SBA-15) were used as heterogeneous catalysts. Furthermore, to allow a comparison between the homogeneous and heterogeneous performance of the most active HPAs, the acidity of the composites was also maintained for 0.373 mmol H^+^/g, and the reactions were performed using the same experimental conditions as before applied for the homogeneous HPAs. The results obtained are displayed in Figure 4, where it is clearly observed that, for both reaction temperatures of 25 and 60 °C, the H_3_PW_12_@*Aptes*SBA-15 exhibits higher glycerol conversion (83% and 91% after 5 min, respectively) than the H_3_PMo_12_@*Aptes*SBA-15 (31% and 40% after 5 min, respectively) composite. Furthermore, H_3_PW_12_@*Aptes*SBA-15 also exhibits the highest solketal selectivity of the two composites, with 97% at both temperatures, 25 and 60 °C. Comparing the catalytic results obtained for the two different temperatures, it is possible to verify that an increase in temperature from 25 to 60 °C resulted in only a slightly better conversion (Figure 4). This indicates that, even at 25 °C, the acetalization reaction of glycerol with acetone, in the presence of HPAs@*Aptes*SBA-15 catalysts, is still a fast-occurring reaction, with the maximum conversion being obtained around 60 min.

The contribution of the functionalized support *Aptes*SBA-15 to the catalytic activity of the composites was investigated. It was used under the same catalytic conditions as previously employed for the composites. However, when using *Aptes*SBA-15, absence of catalytic activity was obtained, meaning that the acidic sites of the HPAs that were incorporated in the composite structure are the catalytic active centers and are responsible for the glycerol conversion to solketal. Such results have been previously reported in the literature, being linked to the low acidic behavior of the SBA-15 support [29,30].

### 3.5. Catalyst Reutilization and Recycling

The most active heterogeneous catalyst, H_3_PW_12_@*Aptes*SBA-15, was selected for catalyst reutilization tests. The reusability capacity of this catalyst was investigated by performing the acetalization reaction of glycerol under the same experimental conditions for various reaction cycles: normalized catalyst acidity of 0.373 mmol H^+^/g, a glycerol/acetone ratio of 1:15 and reaction temperature of 25 and 60 °C. Four consecutive reaction cycles were performed, and after each cycle, the reactional solution (unreacted vestigial glycerol, acetone, and dissolved products) was removed, and a novel amount of glycerol and acetone was added. The catalyst remained intact, without any treatment between cycles. The reutilization performance of H_3_PW_12_@*Apte*sSBA-15 can be observed in Figure 5, for both 25 and 60 °C. At both temperatures, a decrease in catalyst activity can be observed, particularly after the 2nd cycle. The most accentuated drop-in activity is observed at 25 °C, with the catalyst demonstrating 28% of glycerol conversion after the 4th cycle, accompanied by a loss in solketal selectivity from 97% (1st) to 71% (4th). The influence of reaction temperature on the reutilization of HPAs@*Aptes*SBA-15 evaluated at 60 °C shows a smaller loss of catalytic efficiency after the 2nd reaction cycle (69% conversion after the 4th cycle).

Recycling studies, where the catalyst was washed and activated after each cycle, corroborated with the results obtained previously for the reutilization (without catalyst treatment between cycles), with H_3_PW_12_@*Aptes*SBA-15 demonstrating a loss of catalytic activity at 25 and 60 °C (Figure 6). After the 4th cycle at 25 °C, the catalyst is considered inactive (glycerol conversion of 9%). When using the temperature of 60 °C, a smaller loss of catalytic activity was found (54% of conversion after the 4th cycle).

The utilization of the H_3_PW_12_@*Aptes*SBA-15 catalyst in consecutive reaction cycles, with or without cleaning treatment between cycles, originated in a decrease in glycerol conversion after the 2nd cycle. This is probably due to the catalyst suffering a decrease in acidity after the second consecutive reaction cycle. The loss of acidity may be related to the leaching of the HPAs active sites from the *Aptes*SBA-15 support, a phenomenon previously reported in the literature for other catalytic systems [29,31].

### 3.6. Comparison with Other Catalysts

Only a few works can be found in the literature using silica-based catalysts in the acetalization of glycerol with acetone, under a solvent-free system (Table 4). In all the reported examples, the selectivity for solketal was higher than 97%, and most of the reactions were performed at room temperature, highlighting the high suitability of silica-based catalysts for this reaction. Some of these used acid-functionalized silica materials, as shown by Vicent et al., who studied the application of sulfonic acid-functionalized silicas [32]. Among the two catalysts, the Ar-SBA-15 exhibited the highest conversion results, with 82.5% conversion after 0.5 h. Another strategy that has also been applied is the incorporation of different metals into the SBA-15 framework. Ammaji et al. obtained high conversions using an Nb-SBA-15 catalyst (95% of conversion after 1 h), with 100% of solketal selectivity [30]. However, this catalyst showed a loss of catalytic activity in consecutive reusing reaction cycles. The best catalytic performance from HPA@silica catalysts was reported by Gadamsetti et al., where the incorporation of molybdenum phosphate into SBA-15 support resulted in a complete glycerol conversion and very high solketal selectivity (98%) after 1 h of reaction [29]. However, in this work, a loss of active centers from support was also reported in consecutive reactions. Comparing the present work and the results obtained with the H_3_PW_12_@*Aptes*SBA-15 composite (Table 4), identical results were obtained with a shorter reaction time (0.08 h) when the temperature was increased to 60 °C. In future experiments, the ratio of glycerol/acetone will be optimized to decrease the amount of used acetone. Under these conditions, the loss of activity during consecutive cycles can probably be avoided or decreased. However, the loss of activity during consecutive reactions was also observed in most of the reported studies using silica-based catalysts for the acetalization of glycerol with acetone. 

### 3.7. Catalyst Characterization after Use

After four recycling cycles, the H_3_PW_12_@*Aptes*SBA-15 catalyst was recovered and characterized by potentiometric titration, ICP-OES, XRD, and SEM-EDS. The occurrence of HPA leaching and the integrity of the *Aptes*SBA-15 structure were investigated after catalytic use. The heterogeneous catalyst demonstrated a substantial decrease in acidity after the fourth reaction cycle, corroborating with the decrease of the H_3_PW_12_ loading after use (Table 5). The XRD pattern of the spent catalyst (Appendix A) exhibits a loss in intensity of the (100) plane, also hindering peak identification for the (110) and (200) planes. SEM imaging demonstrated the occurrence of particle dispersion while maintaining the characteristic *Aptes*SBA-15 morphology (Appendix A). The presence of H_3_PW_12_ was confirmed by the presence of tungsten in the EDS spectra (Appendix A).

## 4. Conclusions

HPA@*Aptes*SBA-15 composites were prepared, characterized, and further used as heterogeneous catalysts for the valorization of glycerol through acetalization reactions with acetone. The *Aptes*SBA-15 support showed an absence of catalytic activity owing to its characteristic low acidity. On the contrary, the two prepared composites H_3_PW_12_@*Aptes*SBA-15 and H_3_PMo_12_@*Aptes*SBA-15 showed to be active, mainly when the reaction temperature was increased to 60 °C. The H_3_PW_12_@*Aptes*SBA-15 exhibited superior glycerol conversion (91% after only 5 min and 97% after 60 min, with 97% solketal selectivity). This catalyst was used for four consecutive reaction cycles, with and without the cleaning process of the catalyst, between reaction cycles. In both cases, a loss of catalytic efficiency was observed after the second cycle. The reason for this loss was investigated by performing the characterization of the catalyst after catalytic use during four consecutive reactions. By acidity and ICP measurements, it was possible to identify the leaching of the H_3_PW_12_ catalytic centers. On the other hand, the support materials seem to maintain their structure by using XRD and SEM analysis. Comparing the performance of H_3_PW_12_@*Aptes*SBA-15 with other previously reported results in the literature, the most prominent advantage of this prepared catalyst is the high yield of solketal that was obtained after only 5 min of reaction. However, the weakness of this composite lies in the stability of H_3_PW_12_ on its surface. In the near future, different strategies will be adopted to decrease or even eliminate this faintness: decrease the amount of acetone used, functionalize the surface of the silica support with more effective functional groups, and use other structures of silica supports.

## Figures and Tables

**Figure 1 nanomaterials-14-00733-f001:**
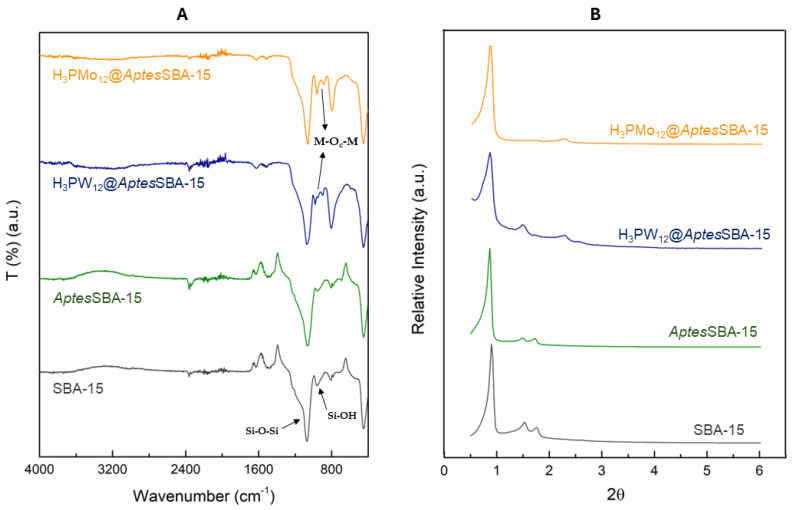
(**A**) FTIR spectra of SBA-15, *Aptes*SBA-15, H_3_PW_12_@*Aptes*SBA-15, and H_3_PMo_12_@*Aptes*SBA-15. (**B**) Powder XRD patterns for SBA-15, *Aptes*SBA-15, H_3_PW_12_@*Aptes*SBA-15, and H_3_PMo_12_@*Aptes*SBA-15.

**Figure 2 nanomaterials-14-00733-f002:**
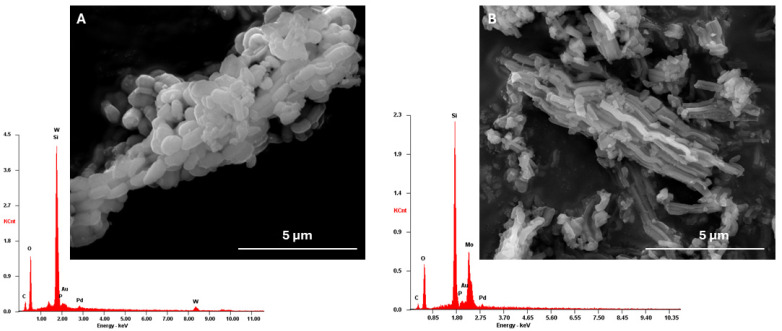
SEM images and EDS spectra obtained for (**A**) H_3_PW_12_@*Aptes*SBA-15 and (**B**) H_3_PMo_12_@*Aptes*SBA-15.

**Figure 3 nanomaterials-14-00733-f003:**
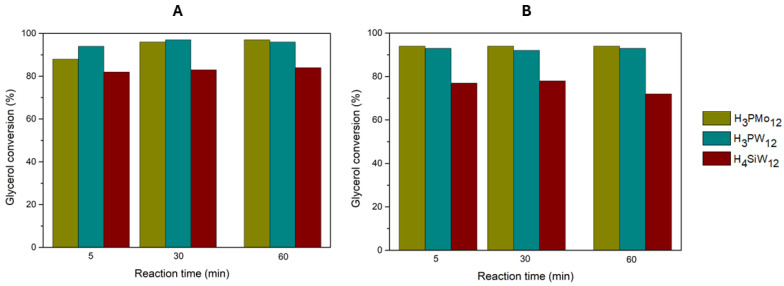
Glycerol conversion results obtained for the three homogeneous catalysts (H_3_PMo_12_, H_3_PW_12_, and H_4_SiW_12_), using a ratio of 1:15 glycerol/acetone, a catalyst acidity of 0.373 mmol H^+^/g, at (**A**) 25 °C and (**B**) 60 °C.

**Figure 4 nanomaterials-14-00733-f004:**
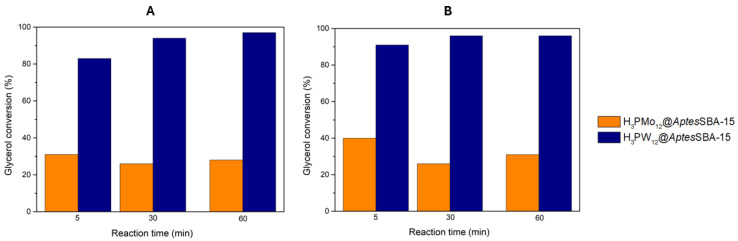
Glycerol conversion results obtained for the two composite catalysts (H_3_PMo_12_@*Aptes*SBA-15 and H_3_PW_12_@*Aptes*SBA-15), using a ratio of 1:15 glycerol/acetone, normalized catalyst acidity of 0.373 mmol H^+^/g, at the reaction temperature of (**A**) 25 °C and (**B**) 60 °C.

**Figure 5 nanomaterials-14-00733-f005:**
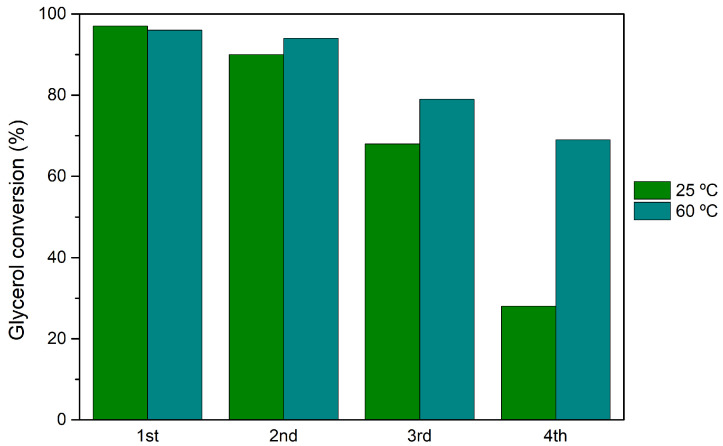
Glycerol conversion results obtained after 1 h of glycerol acetalization with acetone, using H_3_PW_12_@*Aptes*SBA-15 catalyst, after four consecutive reutilization cycles. Reactions performed using a ratio of 1:15 glycerol/acetone, normalized catalyst acidity of 0.373 mmol H^+^/g, at the reaction temperature of 25 and 60 °C.

**Figure 6 nanomaterials-14-00733-f006:**
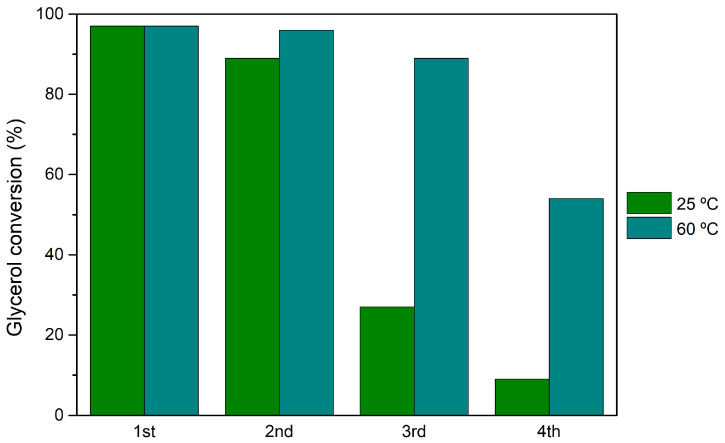
Glycerol conversion results obtained after 1 h of glycerol acetalization with acetone, using H_3_PW_12_@*Aptes*SBA-15 catalyst, after four consecutive recycling cycles. Reactions performed using a ratio of 1:15 glycerol/acetone, normalized catalyst acidity of 0.373 mmol H^+^/g, at the reaction temperature of 25 and 60 °C.

**Table 1 nanomaterials-14-00733-t001:** Textural properties of the HPAs@*Aptes*SBA-15 composites.

Material	S_BET_ (m^2^/g)	V_p_ (cm^3^/g)
H_3_PW_12_@*Aptes*SBA-15	129	0.245
H_3_PMo_12_@*Aptes*SBA-15	188	0.293
*Aptes*SBA-15	337	0.589
SBA-15	725	0.971

**Table 2 nanomaterials-14-00733-t002:** Acidity and pH values for the HPAs and the HPAs@*Aptes*SBA-15 composites.

Material	pH	Acidity (mmol H^+^/g)
H_3_PMo_12_	2.2	9.74
H_3_PW_12_	2.2	1.93
H_4_SiW_12_	2.6	0.995
H_3_PMo_12_@*Aptes*SBA-15	3.41	3.36
H_3_PW_12_@*Aptes*SBA-15	3.39	0.470

**Table 3 nanomaterials-14-00733-t003:** Normalized acidity and corresponding molarity of the HPA catalysts used.

Material	Acidity (mmol H^+^/g)	Catalyst Amount (µmol)
H_3_PMo_12_	0.373	0.525
H_3_PW_12_	0.373	1.70
H_4_SiW_12_	0.373	3.26

**Table 4 nanomaterials-14-00733-t004:** Silica and HPAs@silica composites used as catalysts in the acetalization of glycerol with acetone, in a solvent-free environment.

Catalyst	Ratio of Glycerol/Acetone	T (°C)	Time (h)	Conversion (%)	Selectivity to Solketal (%)	Ref.
SO_4_-Al-MCM-41	1:10	RT	2	94.8	99	[33]
PSF@SiO_2_	1:10	RT	1.5	86.6	98	[34]
Ar-SBA-15	1:6	70	0.5	82.5	wi	[32]
Pr-SBA-15	1:6	70	0.5	79.0	wi	[32]
Nb-SBA-15	1:3	RT	1	95	100	[30]
Zr-SBA-15	1:3	RT	1	92	98	[30]
Ti-SBA-15	1:3	RT	1	65	98	[30]
Al-SBA-15	1:3	RT	1	60	98	[30]
MoPo/SBA-15	1:3	RT	1	100	98	[29]
Cs_2_._5_H_0_._5_PW_12_O_40_@KIT-6	1:6	RT	0.25	95	98	[20]
H_3_PW_12_@*Aptes*SBA-15	1:15	RT	0.08	83	97	This work
		60		91	97	
H_3_PMo_12_@*Aptes*SBA-15	1:15	RT	0.08	31	69	This work
		60		40	76	

wi means without information; RT means room temperature.

**Table 5 nanomaterials-14-00733-t005:** Acidity of H_3_PW_12_@*Aptes*SBA-15 composite, obtained through potentiometric titration and ICP-OES analysis before use and after its utilization during four catalytic cycles.

H_3_PW_12_@*Aptes*SBA-15	Acidity (mmol H^+^/g)	HPA Loading (µmol/g)
Before usage	0.469	159
after 4 catalytic cycles	0.277	105

## Data Availability

The data presented in this study are available on request from the corresponding author.

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
