# Peer review of "Heteropolyacids@Silica Heterogeneous Catalysts to Produce Solketal from Glycerol Acetalization"

_nanomaterials, 2024, doi:10.3390/nano14090733_

Round 1
Reviewer 1 Report
Comments and Suggestions for Authors
In the context of biorefineries, the production of solketal from two molecules of biomass is a reaction that has been widely studied in the literature and is promising. Therefore, obtaining active, selective and stable catalysts is a topic of interest in catalysis and chemical engineering.
The paper is well written and the results are clearly presented. The catalytic results obtained are good, especially with the H3PW12@AptesSBA-15 catalyst. The work has potential, but some points could be improved before acceptance.
1) The reaction studied is exothermic. Could the authors explain the conversion results obtained as a function of temperature?
2) The catalysts were prepared by drying at RT. For use at 60°C, do the catalytic systems have the same number of acid sites? Some quantification of the total number of sites present in the post-reaction catalysts would contribute to this work.
3) These systems also contain Lewis acid sites. What is their role in activity and selectivity?
4) Could the authors show the selectivity achieved under all conditions? Also, what by-products were obtained?
Author Response
The authors acknowledge the Referees helpful and relevant comments. They have received careful consideration on our part and have resulted in the introduction of some revisions in the manuscript that are highlighted in yellow in response to their requests, as follows:
In the context of biorefineries, the production of solketal from two molecules of biomass is a reaction that has been widely studied in the literature and is promising. Therefore, obtaining active, selective and stable catalysts is a topic of interest in catalysis and chemical engineering.
The paper is well written and the results are clearly presented. The catalytic results obtained are good, especially with the H3PW12@AptesSBA-15 catalyst. The work has potential, but some points could be improved before acceptance.
- The reaction studied is exothermic. Could the authors explain the conversion results obtained as a function of temperature?
Authors Answer: Using the homogeneous catalysts only slightly difference of solketal conversion was observed when the reaction temperature was increased from the room temperature to 60 ºC (Figure 3). Similar behaviour was found for the heterogeneous composites (Figure4). The influence of reaction temperature is only observed for the reutilization and reusing studies using HPAS@SBA-15 catalysts, where smaller loss of catalytic activity was found using 60ºC instead of 25 ºC. As described in the manuscript, a loss of catalyst acidity must occur during consecutive reactions cycles, and higher reaction temperature must promote a higher activity of a lower acidic catalysts, probably caused by a higher diffusion of reactants using 60ª instead of room temperature.
- The catalysts were prepared by drying at RT. For use at 60°C, do the catalytic systems have the same number of acid sites? Some quantification of the total number of sites present in the post-reaction catalysts would contribute to this work.
Authors Answer: Yes, the same catalyst acidity was used for the reaction at RT and at 60 ºC. Table 5 presents the results of acidity before and after catalytic usage during four catalytic cycles, where it is possible to confirm the loss of acidity from 0.469 mmol H+/g to 0.277 mmol H+/g. Also, by ICP was possible to confirm the loss of HPA from HPA@SBA-15 composite.
- These systems also contain Lewis acid sites. What is their role in activity and selectivity?
Authors Answer: HPAs catalysts possess both Brönsted and Lewis acid sites. Both may have an important role in the conversion of glycerol and solketal selectivity. However, it is well described in the literature that the Bronstead acid centres are more important to achieve faster high yield of Solketal (doi.org/10.1021/acs.jpcc.6b04229; PERSPECTIVA, Erechim. v. 41, n.155, p. 101-112, setembro/2017). It is mentioned that Lewis acid sites have lower capacity to convert the acetone−glycerol adduct into solketal, as compared to the Brønsted acid sites (doi.org/10.1021/acs.jpcc.6b04229). Therefore, the formation of the acetone−glycerol adduct is not considered the rate-determining step, but its subsequent conversion into solketal. Brønsted acid sites are far more efficient than Lewis ones to convert the acetone−glycerol adduct into solketal.
- Could the authors show the selectivity achieved under all conditions? Also, what by-products were obtained?.
Authors Answer: As described in page 5, line 180, “Solketal selectivity was above 97% for all Catalysts”. Therefore, in all reactions performed practically only solketal was obtained in all reaction performed, and only vestigial 2,2-dimethyl-1,3-dioxan-5-ol amount was obtained. This information was added to the manuscript.
Reviewer 2 Report
Comments and Suggestions for Authors
In this manuscript, hybrid catalysts consist of heteropolyacids (H3PW12, H3PMo12) with amine-functionalized silica (AptesSBA-15) materials were developed. The functional group, morphology, porous structure and acidity of HPAs and HPAs@AptesSBA-15 were discussed. Among these catalysts, the H3PW12@AptesSBA-15 exhibited 91% glycerol conversion after only 5 min with 97% solketal selectivity. However, the catalysts became almost inactive after 4th cycle at 25℃ related to the leaching of the HPAs active sites from the AptesSBA-15 support. Overall, I believe it can be accepted by Nanomaterials after major revision. The following is the specific comments:
1. As for the FTIR spectra in the Figure 1., although the C−N stretching vibration overlay with the IR absorptions of Si−O−Si and of Si−CH2−R, I would suggest the author to give more discussion to clarify the amine group on the functionalized silica, such as the N-H binding in the range of 1600-170 cm-1, which should be a typical signal of amine group.
2. What is the purpose to used amine-functionalized silica instead of pristine silica as the support? In other words, does the amine group have some interaction effect with HPA? And control experiment of pristine silica is suggested to be added to the main session.
3. The authors claim that "These suggested that the HPAs have higher acidity when compared to the corresponding HPAs@AptesSBA-15 composites. "(Line 146-147) Please give more explanation about these results.
4. The authors observe that "After the 4th cycle at 25 °C, the catalyst was considered inactive (glycerol conversion of 9%). Using the temperature of 60 °C, smaller loss of catalytic activity was found (54% of conversion, after the 4th cycle)." (Line 263-264). In my understanding, higher temperature could result more HPA leaching out from the support, so that give a more decreased activity. Could the author explain these funding?
5. Some small points: 1) It would be more readable if the functional peaks could be marked on the Figure 1., 2) Nitrogen adsorption desorption isotherms of pristine AptesSBA-15 were suggested to be added.
Author Response
Response to reviewer 2
The authors acknowledge the Referees helpful and relevant comments. They have received careful consideration on our part and have resulted in the introduction of some revisions in the manuscript that are highlighted in green in response to their requests, as follows:
In this manuscript, hybrid catalysts consist of heteropolyacids (H3PW12, H3PMo12) with amine-functionalized silica (AptesSBA-15) materials were developed. The functional group, morphology, porous structure and acidity of HPAs and HPAs@AptesSBA-15 were discussed. Among these catalysts, the H3PW12@AptesSBA-15 exhibited 91% glycerol conversion after only 5 min with 97% solketal selectivity. However, the catalysts became almost inactive after 4th cycle at 25℃ related to the leaching of the HPAs active sites from the AptesSBA-15 support. Overall, I believe it can be accepted by Nanomaterials after major revision. The following is the specific comments:
- As for the FTIR spectra in the Figure 1., although the C−N stretching vibration overlay with the IR absorptions of Si−O−Si and of Si−CH2−R, I would suggest the author to give more discussion to clarify the amine group on the functionalized silica, such as the N-H binding in the range of 1600-1700 cm-1, which should be a typical signal of amine group..
Authors Answer: The authors acknowledge the reviewer for this important point. It is well reported in the literature that the functionalization of SBA-15 with APTES can present in its FTIR spectrum a characteristic band around 1540 cm-1, which is attributed to N–H stretching vibrations of aminopropyl anchored on the surface of mesoporous support. However, the intensity of this band is week and when the presence of APTES functional groups in not largely enough, this N-H band cannot be observed in the FTIR spectrum of functionalized silica APTES-SBA-15 and composites. This information was added to the manuscript, and it is highlighted in green.
- What is the purpose to used amine-functionalized silica instead of pristine silica as the support? In other words, does the amine group have some interaction effect with HPA? And control experiment of pristine silica is suggested to be added to the main session.
Authors Answer: The authors acknowledge the reviewer for this important point. The interaction of HPAs and non-functionalized SBA-15 is week and a large leaching of HPAs occurs, mainly in polar reaction media. The literature presents several success examples in the immobilization of HPAs on an amine functionalized silica surface. Therefore, an efficient method to improve the immobilization of HPAs active homogeneous catalysts, is the creation of anchoring points in the surface of the support. In this case, an electrostatic interaction of HPAs occurs with the aptes groups in the surface of SBA-15. This information was introduced in the manuscript in page 2, line 74 and it is highlighted in green.
- The authors claim that "These suggested that the HPAs have higher acidity when compared to the corresponding HPAs@AptesSBA-15 composites. "(Line 146-147) Please give more explanation about these results.
Authors Answer: Once again, authors acknowledge the review for this important point. This was better clarified in the manuscript since the comparison performed was for 1 gram of material. This correction was present in section 2.2, and it is highlighted in green.
- The authors observe that "After the 4th cycle at 25 °C, the catalyst was considered inactive (glycerol conversion of 9%). Using the temperature of 60 °C, smaller loss of catalytic activity was found (54% of conversion, after the 4th cycle)." (Line 263-264). In my understanding, higher temperature could result more HPA leaching out from the support, so that give a more decreased activity. Could the author explain these funding?
Authors Answer: The authors agree with the reviewer. In fact, a higher reaction temperature can result in a higher leaching process. On the other hand, when diffusion limitation could be present, the increase of reaction temperature can help to decrease the diffusion obstacles. In the case of using heterogeneous porous catalysts HPAs@aptesSBA-15, the occurrence of diffusion difficulty of viscose glycerol medium can be diminished by increasing temperature.
- Some small points: 1) It would be more readable if the functional peaks could be marked on the Figure 1., 2) Nitrogen adsorption desorption isotherms of pristine AptesSBA-15 were suggested to be added.
Authors Answer: The authors acknowledge the reviewer for this important point. The N2 adsorption-desorption data of the SBA-15 support, and its functionalized aptesSBA-14 materials was introduced in the manuscript in page 3 and it is highlighted in green. Peaks identification was introduced in figure 1.
Round 2
Reviewer 1 Report
Comments and Suggestions for Authors
The paper was reviewed and can be accepted in its present form.
Author Response
The authors acknowledge the helpful feedback from the referees and the suggested minor editing of the English language required. These suggestions have been carefully considered on our part and have resulted in the introduction of some revisions in the manuscript, which are highlighted in blue.

Reviewer 2 Report
Comments and Suggestions for Authors
This version is good.
Author Response

(The authors gave the same response as above.)
